# The COMMAND trial of cognitive therapy for harmful compliance with command hallucinations (CTCH): a qualitative study of acceptability and tolerability in the UK

Max Birchwood,[1] Laura Mohan,[2] Alan Meaden,[3] Nick Tarrier,[4] Shon Lewis,[4] Til Wykes,[5] Linda M Davies,[6] Graham Dunn,[7] Emmanuelle Peters,[5] Maria Michail[8]

For numbered affiliations see end of article.

**Correspondence to**
Professor Max Birchwood;
m.j.birchwood@warwick.ac.uk

## ABSTRACT

**Objectives** To explore service user experiences of a 9-month cognitive behavioural therapy for command hallucinations in the context of a randomised controlled trial including their views on acceptability and tolerability of the intervention.

**Design** Qualitative study using semistructured interviews.

**Setting** The study took place across three sites: Birmingham, Manchester and London. Interviews were carried out at the sites where therapy took place which included service bases and participants' homes.

**Participants** Of 197 patients who consented to the trial, 98 received the Cognitive Behavior Therapy for Command Hallucinations (CTCH) intervention; 25 (15 males) of whom were randomly selected and consented to the qualitative study. The mean age of the sample was 42 years, and 68% were white British.

**Results** Two superordinate themes were identified: participants' views about the aspects of CTCH they found most helpful; and participants' concerns with therapy. Helpful aspects of the therapy included gaining control over the voices, challenging the power and omniscience of the voices, following a structured approach, normalisation and mainstreaming of the experience of voices, and having peer support alongside the therapy. Concerns with the therapy included anxiety about completing CTCH tasks, fear of talking back to voices, the need for follow-up and ongoing support and concerns with adaptability of the therapy.

**Conclusions** Interpretation: CTCH was generally well received and the narratives validated the overall approach. Participants did not find it an easy therapy to undertake as they were challenging a persecutor they believed had great power to harm; many were concerned, anxious and occasionally disappointed that the voices did not disappear altogether. The trusting relationship with the therapist was crucial. The need for continued support was expressed.

**Trial registration number** ISRCTN62304114, Pre-results.

### Strengths and limitations of this study

► The study was presented in line with the Consolidated Criteria for Reporting Qualitative Research ensuring the explicit and comprehensive reporting of all study elements.

► Interviews and analyses were conducted by team members not involved in the delivery of the intervention or data collection for the larger randomised controlled trial. However, it is not possible to rule out social desirability bias.

► The absence of participants who did not engage in therapy at all or dropped out as their views would have been valuable in highlighting difficulties with engagement and reasons for withdrawing from Cognitive Behavior Therapy for Command Hallucinations.

the individual and for other people and is a major cause of clinical and public concern. Many individuals who experience auditory hallucinations include commands that instruct them to do harm to self, others or objects.[1 2] Up to half of the adult patients with psychiatric disorders continue to hear voices; of these, 48% stipulate harmful or dangerous actions rising to 69% among patients in medium secure units.[1] While it is difficult to predict individual acts of compliance, those who have complied with voices with serious harm to themselves or others within the previous 9 months are at high risk of repeating this in the following 18 months.[3 4]

Using the framework of our cognitive model of voices, we have shown that it is not only the level of activity of voices, or indeed their content that drives affect and behaviour, but the nature of the relationship with the personified voice.[5–7] We showed that where the voice-hearer believes the voice to

## BACKGROUND

Acting on command hallucinations in psychosis can have serious consequences for

have malevolent intent, and crucially to have the power to deliver the threat, this can motivate compliance or appeasement behaviour.[8] These findings have been independently replicated in a forensic population.[9] This theoretical framework informed the development of our Cognitive Behavior Therapy for Command Hallucinations (CTCH) designed to weaken and challenge beliefs about voices' power, thus enabling the individual to break free of the need to comply or appease and thereby reduce harmful compliance behaviour and distress.[10 11] CTCH is administered over a maximum period of 9 months which includes a therapeutic window up to 25 sessions. The primary aim of the therapy is to reduce the power differential between voice and voice-hearer and to test out the perceived power of the voice by examining evidence for: (a) the voice-hearer's perceived lack of control over voice activity, (b) the perceived omniscience of the voice (eg, the apparent ability of the voice to predict the future) and (c) the perceived capacity of the voice to carry out its threats for non-compliance.

We completed the first major trial of CTCH focusing on this dimension of the voice experience, the perceived power of the voice, to reduce the motivation to comply.[12] The trial demonstrated that the risk of further serious compliance was high in the treatment as usual group (46%), reducing to 28% in those receiving the CTCH. The rate of consent to the trial and adherence to the CTCH were in excess of 80%, providing indirect evidence that the therapy was needed and was acceptable.

The aim of this study was to explore service user experiences of a 9-month CTCH in the context of a randomised controlled trial (RCT; ISRCTN62304114) including their views on acceptability and tolerability of the intervention.

## METHODS
### Study design and settings
This was a qualitative study embedded within a larger RCT comparing CTCH and treatment as usual with treatment as usual alone. The trial involved 197 eligible participants from three UK centres. Participants were assigned to the intervention or control arm of the trial in a 1:1 ratio. The treatment period was a maximum of 9 months and follow-up occurred at 9 and 18 months after randomisation. Participants were eligible if they met all of the following criteria: had International Statistical Classification of Diseases and Related Health Problems, 10th Revision, schizophrenia, schizoaffective (F20, 22, 23, 25, 28, 29) or mood disorders (F19 32) and were under the care of a clinical team; were aged 16 years and older; had a history of harmful command hallucinations for at least 6 months with recent (<9 months) history of harm to self or others, or major social transgressions as a result of the commands (full or incomplete compliance); or had harmful command hallucinations whereby the individual was distressed and appeasing the powerful voice. Individuals with a primary diagnosis of addictive disorder, an organic

impairment or insufficient command of English were excluded from the study. The protocol for the cognitive therapy, described in Meaden et al[10] allows the treatment to be tailored to the individual. The intervention was described in the main trial publication, with illustrative cases written up in our manual.[10 11]

### Research team and reflexivity
The interviews were conducted by one of the coapplicants (Helen Lester, deceased) and two research assistants, all unknown to participants, and examined the thoughts, concerns and experiences of the CTCH in an open manner. Therapists did not conduct the interviews nor were they present during interviews. The interviews continued until the participant felt they had nothing more to say in response to the framework questions proposed to them (probe questions). The interviews each lasted for approximately half an hour. Interviews were coded by two raters not involved in the trial. We believe therefore that the data collected here were representative of those receiving therapy and interpreted without bias.

### Patient and public involvement
Our progenitor paper[3] interviewed patients about what was troublesome about voices and what they wanted to change; command hallucinations emerged as a frequent issue. The intervention was developed in 2006[11] in conjunction with patients, who were consulted on outcomes and process. This paper further assesses patients' views on the therapy including the burden of the intervention. A service user in research group based in Birmingham (SURESEARCH) was consulted on the research question when the study was designed (2005) and a nominee invited to the trial steering committee. Patients were not involved in recruitment or conduct of the study. Participants were not informed directly of the trial outcome but were able to access the open access primary publication.

### Sample
A sample of 25 participants was drawn, who were selected randomly by number generator, from the total of 98 receiving the therapy in the trial (25.5%). These included 8 from the Birmingham site, 10 from London and 7 from Manchester, with a range of ages and genders included. The sample included participants who had completed all of the sessions as part of the CTCH programme, and who consented to be interviewed and recorded for the purposes of this study. Interviews were carried out until theoretical saturation was achieved with no new themes emerging. This was achieved at 25 interviews. Participants were assured that the interviews would be anonymous, and individual views of therapists would not be communicated to the therapists.

### Data collection and analysis
Face-to-face semistructured qualitative interviews were carried out at the sites where therapy took place which

included service bases and participants' homes. Each interview was audio-recorded, transcribed verbatim and checked for accuracy. Personal details were removed to preserve participant anonymity. Field notes were written in the form of memos. The study is presented in line with the Consolidated Criteria for Reporting Qualitative Research (COREQ)[13] and the Standards Reporting Qualitative Research.[14]

The interviews were structured but flexible in allowing follow-up to participant-supplied information. They focused on a brief history of the patients' difficulties and symptoms, followed by discussion of the therapy they had received, and the positive and negative aspects of receiving the therapy as felt by the participants. The interview guide included the following probe questions:

*Do you feel that the therapy helped you? Why do you feel that, that session (which participant mentioned) was (un)helpful for you?*

*Was the therapy acceptable for you?*

*How was your relationship with (the therapist)?*

*Did you feel the therapist understood what it was like to hear voices?*

*How was the length of the sessions and the programme as a whole?*

*What was the most helpful session? What was the least helpful session?*

*Do you feel the therapy made a difference at all?*

The interviews were analysed using adapted grounded theory.[15] Theories were generated from the data. Evidence of dissonance was sought throughout, and theories were altered to accommodate this. Quotations have been chosen to be representative of the data collected. Each transcript was read and re-read by LM and Elizabeth England (EE), until saturation of data was reached, and no new themes emerged.

There were line-by-line thematic analysis and discussion on any discourse present in the analysis of the themes. Words, ideas and reported experiences in the participants' texts was completed with an inductive approach with no pre-existing framework for the ideas. Agreement between LM and EB was high when it was discussed at regular meetings. Thematic coding was grouped into key emergent themes, which we present as findings in this paper.

## FINDINGS

The sample consisted of 25 participants, 15 of whom were males. The mean age of the sample was 42 years, and 68% were white British. Thirty-six per cent of the sample were diagnosed as 'unspecified psychosis', 56% as 'schizophrenia' and 8% as schizo-affective disorder. The demographic characteristics of the sample are presented in table 1.

Two superordinate themes were identified: participants' views about the aspects of CTCH they found most helpful and participants' concerns with therapy. The themes (and subthemes) are presented below and supported by illustrative quotations from the transcripts.

## Helpful aspects of CTCH

The components of CTCH participants reported finding most helpful in tackling their mental health problems were related to:

### Feeling more in control

Several individuals discussed one of the outcomes of the therapy as gaining more control over the voices and being able to cope better with their mental health problems and the impact these had on their lives. Gaining control over the voices was one of the main components and aims of the intervention, and in our previous work we were able to show that participants who received the CTCH intervention felt more powerful, than those in the control group, in their ability to withstand and mitigate the threat received by their voices.

> I'm able to control them [the voices] a bit more. (F52BIR104)

> I cope better now because I listen to what [the therapist] said. (F67LON013)

> I feel like I've got control of them [the voices]. (M30BIR125)

### Challenging the power and omniscience of the voices

Participants spoke about how being encouraged to reflect on evidence for voices' power and omniscience particularly helped them during the therapy. Using examples from everyday life, therapists and participants worked collaboratively to show that voices made empty threats which subsequently led participants to realise that '*the problems are all inside me. They are not out there*', as one participant noted. The therapy aimed to build on participants' own strengths to face up to the voices, and so make the voices less fearful. As indicated below, this technique was reported as particularly useful by the participants.

> She said, if we put the papers down and let's see if they come in and take them, and I said they didn't. [In response to patients belief that the voices would take her therapy notes from the therapist to find out what she had told others of the voices] I think that helped, you know. It just gives me a bit of support that they're lying. They are just talking. (F67LON013)

> She [the therapist] was pregnant and [the voice] says that something will happen to her baby. And she had the baby and nothing happened, and that was really a good reassurance that [the voice] is just in my mind and not real. [The therapist] says he [the voice] could bring it on, that it won't hurt her and she was sure the baby would be alright. I know he can't do things physically, just mentally. (F57BIR094)

> [The voice] is just in my mind and not real. (F57BIR094)

### Following a structured approach

Participants described how taking a structured approach to the therapy sessions helped them get an overview of

**Table 1** Participant characteristics

| Identifier | Gender | Age | Ethnicity | Diagnosis |
|---|---|---|---|---|
| F56LON009 | Female | 56 | British | Unspecified psychosis |
| F67LON013 | Female | 67 | British | Unspecified psychosis |
| M53LON072 | Male | 53 | Other—white background | Schizophrenia |
| F42LON077 | Female | 42 | Caribbean | Schizo-affective disorder |
| F48LON085 | Female | 48 | British | Unspecified psychosis |
| M18LON099 | Male | 18 | British | Unspecified psychosis |
| F48LON100 | Female | 48 | British | Unspecified psychosis |
| M41LON101 | Male | 41 | British | Schizophrenia |
| F27LON105 | Female | 27 | British | Schizophrenia |
| M53LON001 | Male | 53 | Other—black background | Schizophrenia |
| M39BIR093 | Male | 39 | British | Schizophrenia |
| F57BIR094 | Female | 57 | British | Schizophrenia |
| M55BIR016 | Male | 55 | Irish | Schizophrenia |
| M30BIR125 | Male | 30 | Caribbean | Unspecified psychosis |
| M35BIR126 | Male | 35 | British | Unspecified psychosis |
| M37BIR013 | Male | 37 | Caribbean | Schizophrenia |
| F52BIR104 | Female | 52 | British | Schizo-affective disorder |
| M34BIR045 | Male | 34 | British | Schizophrenia |
| F25MAN067 | Female | 25 | British | Unspecified psychosis |
| M25MAN062 | Male | 25 | British | Schizophrenia |
| M31MAN066 | Male | 31 | British | Unspecified psychosis |
| M46MAN076 | Male | 46 | British | Schizophrenia |
| M47MAN073 | Male | 47 | Other—black background | Schizophrenia |
| F40MAN079 | Female | 40 | African | Schizophrenia |
| M50MAN081 | Male | 50 | British | Schizophrenia |

the therapy in the context of a complex and confusing experience. Some participants specifically commented that understanding the triggers for their voices helped to gain mastery, as illustrated by the quotes below.

It was very structured; all the time we had an agenda which was good (F42LON077)

She gave me some information afterwards…I could remember the session where she was helping me through this. (F48LON085)

Connecting things together…change your pattern and change your ways. (F56LON009)

She drew me a diagram between the bad thoughts and the good thoughts, and how the brain works and that diagram was helpful. (F48LON085)

### Normalisation and mainstreaming of the experience of voices

Many participants described how the acceptance of voice hearing reduced feelings of isolation and demonstrated through the experience of others that control was possible. Normalisation of the voice appeared to reduce the 'specialness' of the voice and offered support to the voice-hearer. Participants commented how finding out

that hearing voices is common and '*can happen to most people*' (M25MAN062) was empowering and helped them feel better about themselves.

[The therapist] was like 'oh it's common and can happen to most people.' I wasn't aware of that… She was telling me how the voices can be controlling and stuff. (M25MAN062)

[watched a DVD called 'am I normal' by the BBC, suggested by therapist] people that either hear voices or have heard voices and they're back in work, or they're living some kind of normal life or they're in respected positions, but they still hear voices. That was helpful; let me feel that I wasn't alone. I felt a little bit better in myself. (M53LON001)

### The value of peer support

Participants highlighted that provision of peer support alongside the therapy was beneficial in normalising their experiences, supporting them with challenging or high-risk situations such as going on assignments outside, and encouraging them in engaging with difficult aspects of the therapy such as challenging the power of voices. In

particular, participants felt that peer support helped them feel included and listened to; empowered them to help others, or as one participant said 'give something back'; and also benefited their social life.

> The family and people where I'm living now in support housing [listen to me]. We talk about the voices. They listen to me and talk to me like, saying things aren't going to happen while you're living here. (M30BIR126)

> I'm going to a place on Tuesday, and you talk about the issues you wanna talk about. They're there to listen. (M41LON101)

> She [the therapist] put me in touch with the healing arts team, and another place called [an arts based support group for people with mental health issues] …a couple of times we went outside. (M53LON072)

> [Patient in conversation with another patient with psychosis] I just told her it gets better and he won't win. Even though you won't feel like it now, he won't win. [Interviewer asks if she recommended speaking back to the voices, patient responds] 'yeah tell them to get lost. (F25MAN067)

> I really wanna do something, like give back what they give you. Take it and help other people. (F40MAN079)

> I [the participant] gave her [the therapist] an example on how to cope with voices. (M37BIR013)

### Concerns with therapy

Participants also expressed several concerns with the therapy. These were mainly associated with the experience of persecutory feelings interfering with the therapy process, for example, concerns about the therapist and their intentions, but also receptiveness to accepting psychological therapy. Adapting the therapy to the individual and reassurance regarding the lack of real world power of the voices were reported as helping with the tolerability of the therapy.

### Fear of therapy

Many participants described fear of the therapy arising from a perceived lack of ability to perform the tasks requested as part of the therapy, such as talking back to the voices. This could be related to lack of self-confidence which seemed to be more an issue at the early stages of the therapy yet caused significant distress to participants. A commonly reported concern was talking about the voices to the therapist due to fear about what the voices might do or say. For some individuals, the voices spoke directly about the therapist and instructed the individual not to reveal their conversations.

> I was frightened about starting the therapy, because I didn't know quite what to expect… [During the therapy] I felt unsure whether I was answering them [the questions] correctly. (F48LON085)

> They tell you not to talk about them…so you're frightened to talk to someone about it. (F48LON100)

> It's quite frightening to try and explain these voices. (F67LON013)

### Challenges to established patterns of thinking and behaviour

Some participants described feeling upset at times when asked to think through and challenge long-established beliefs about their voices and their behaviour in response to the voices. For example, one participant described finding it very difficult to talk back to the voices following the therapist's suggestion. Trying out new ways of coping, for example, testing the voice as a way of challenging participants' entrenched beliefs about the voices and their associated power was also something participants found challenging. In some cases, there was resistance to some of the new suggestions and ways of coping. One participant said:

> I got my own way of coping with things and she was trying to introduce new ways and I found it hard to try to practice them coz I was used to thinking a certain way. (M50MAN081)

This resistance could be related to perceived lack of self-confidence and the fear of therapy discussed previously but also highlights other concerns such as participants' fear at changing the strategies they already have for dealing with the voices. Personalising aspects of the therapy to address such concerns would be necessary, as therapy needs to take into account an individual's beliefs, fears and coping methods to plan a way forward.

> It was upsetting at times, I would leave here in floods of tears at times, but I carried on. (F48LON085)

> [the therapist suggested the patient] should talk back to them, try and reason with them, but I didn't think that was a very good idea because then I'm being pulled into their games. (F27LON105)

### Building trust with the therapist

Many participants described how building a trusting relationship with the therapist was difficult as new and sometimes challenging ideas were discussed during therapy which some individuals felt placed them at risk from the voices. Building trust with the therapist was perceived by participants to be important in allowing them to feel confident to change established patterns of thinking and behaviour. Participants explained how this did not happen overnight, and a collaborative effort from the therapist and individual was necessary to achieve that. Some described how their therapist worked practically alongside them in a facilitating role, while others discussed having the therapy in an environment where they felt safe.

> It took me a little while to build up the trust [with the therapist]. (F48LON100).

> I was very frightened about leaving the hospital and on just one occasion she came and took the bus with me part way. She walked me to the bus and got on the bus with me, and I found that very supportive, and it

gave me the encouragement to be able to leave the session and do it on my own. (F42LON077).

She [the therapist] helped me start going places where there's other people. (F25MAN067)

And then we started going [to have the therapy at] my Grans and it was alright… Because I feel safe at my Grans. (F25MAN067)

### Need for better personalisation of therapy

While some participants were pleased that the therapy and the therapists were sensitive to the individual and their circumstances, including any fears and concerns, others voiced concerns that the therapy was not adaptable enough and provided some examples of how the content, duration and tasks of the therapy could be tailored to the individual's needs and preferences.

It [the therapy] should have gone to a practical therapy. Actually confront the fears together and she could go through what I was thinking at certain points. Actually go to the public places. (M50MAN081).

[in response to is there anything you would change] The sessions should last longer sometimes. Or shorter sometimes. [So be more flexible?] Yeah. It was how much it applied to you. (M46MAN076)

She gave me things to do, like do you have any hobbies or something to take your mind off… I was looking at a book I just couldn't read… She said to make some noise like 'mmmmh' to like, to stop it…I felt silly doing that in the living room and my parents were [enquiring why the patient was making the noise] (M25MAN062)

### Expectations of therapy: control versus cure

Participants' expectations of therapy, that is, their beliefs about the consequences of receiving treatment, such as that the treatment will lead to improvement, was a recurrent issue. Some participants reported being disappointed that the therapy did not have the impact they had expected, for example, it did not make the voices go away or as one participant said '*the magic never seemed to start*'.

I just kept thinking, when's the magic going to start. (M50MAN081)

They've [the voices have] not gone away completely. You know unfortunately I thought they would but they haven't. (F42LON077)

### DISCUSSION

In this study, we explored service user experiences of a 9-month cognitive behavioural therapy for command hallucinations in the context of an RCT including their views on acceptability and tolerability of the intervention.

The study was presented in line with COREQ ensuring the comprehensive reporting of all study elements. Interviews and analyses were conducted by team members not involved in the delivery of the intervention or data collection for the larger RCT. Indeed, therapists were not involved in the qualitative study nor were they present during the interviews. A limitation of this study was the absence of participants who did not engage in therapy at all or dropped out as their views would be valuable in highlighting difficulties with engagement and reasons for withdrawing from CTCH.

Few CBT studies in psychosis have sought systematically to document participants' experience of therapy.[16 17] A recent synthesis of existing qualitative studies concluded that an alliance between service user and therapist, clear steps in facilitating change and the challenges of applying cognitive behavioural therapy for psychosis are the main reported experiences of users.[17] In this study too, participants highlighted the importance of the therapist–service user relationship in distancing from voices' power, the value of structure in the therapy and how the CTCH could be difficult and distressing for participants. In addition, the present study also documented how normalising the voice experience served to gain intellectual mastery of it (ie, voices were no longer 'special' or 'mysterious') and many highlighted the need for longer-term, preferably peer support.

The components of the CTCH service users found most useful was gaining more control over the voices and challenging the power of voices to act on their supposed malevolent intent. The words 'control' and 'cope' cropped up frequently in the narratives of participants. Many valued the structured approach to examining long-held beliefs and practices; for example, clarity about the triggers for voices helped to achieve intellectual mastery of the experience and as a basis to challenge the power of the persecutor behind the voice. Others described the value in normalising the experience; this characterisation as an everyday event served as an antidote to the 'special' place of the voice and indeed of the voice-hearer within it. Some argued that this could have gone further with improved peer support and validation. Trusting the therapist was brought up in many of the narratives. This is understandable from the client's point of view as from their perspective, the voice is embodied with major threats, in some cases life-threatening, and hence trusting the therapist's motives and ability was seen as crucial. Participants recounted many examples of the lengths therapists went to achieve this; accompanying them on assignments in supposed high-risk situations was often described and valued by the participants.

Some concerns with the therapy, although minor, were also expressed, and these centred on the experience of persecutory feelings interfering with the therapy process, for example, concerns about the therapist and their intentions but also, receptiveness to accepting psychological therapy. Many participants reported feeling fearful of the therapy and found the process of facing up to the power of voices at times distressing. Simply articulating these complex experiences was sometimes profoundly difficult for individuals. In some

cases, the voices attempted to exert counter-control by questioning the therapist and commanding the individual to disengage. This is perhaps where the suggestion for more peer support from participants would have been beneficial. It was also noted to be beneficial for the therapist to provide reassurance in the form of reiterating that the voices cannot cause affect in the real world. Some participants expressed concerns that the therapy '*did not get rid of the voices'*. Managing expectations of the therapy is crucial for therapeutic success and may influence the relationship between the therapist and the client.[18] Therapists have a key role in communicating clearly at the outset—and throughout the therapy—the intended changes and benefits of the therapy. This will allow a shared understanding of how therapy will proceed, facilitating engagement and rapport. Nonetheless, none of the individuals we spoke to expressed strong adverse experiences of the therapy and as we highlight, few dropped out (n=7).

## Implications for practice

The views of participants captured in this study combined with the high level of completion of the therapy and retention in the trial, indicate that the CTCH is an acceptable intervention and generally well tolerated. Clinicians working with individuals experiencing harmful voices should be mindful of asking, as part of their assessment, about the perceived power of the voice relative to the individual. This is important as our work[19] has shown that the perceived power of voices to threaten the individual was the best predictor of harmful compliance to voices.

Nonetheless, clinicians should also be mindful of challenges reported by individuals in relation to the CTCH, for example, not being an easy therapy to undertake, fear that the approach could increase the perceived threat from voices and maximise efforts to build alliance. We recommend peer-support as a useful addition to this therapy, particularly in providing support following its conclusion but also in providing help during difficult periods when the voice's power is confronted and to further normalise the experience (an approach recently highlighted as a '*promising practice'* in the 2014 National Institute for Health and Care Excellence guidance for schizophrenia).[20] We also recommend to therapists implementing this approach to make it clear that the intervention is not designed to eliminate voices, rather it is to build on individuals' own strengths to face up to the power of the voice, to make them less fearful and place them in the 'driving seat'.

## Author affiliations
¹Division of Mental Health and Wellbeing, Warwick Medical School, University of Warwick, Coventry, UK
²College of Medicine and Dentistry, University of Birmingham, Birmingham, UK
³Faculty of Health Science, Solihull Mental Health NHS Foundation Trust, Birmingham City University, Birmingham, UK
⁴Division of Psychology & Mental Health, University of Manchester, Manchester, UK
⁵Institute of Psychiatry, Kings College, University of London, London, UK
⁶Institute of Population Health, University of Manchester, Manchester, UK
⁷Health Methodology Research, University of Manchester, Manchester, UK
⁸School of Psychology, Institute of Mental health, University of Birmingham, Birmingham, UK

**Acknowledgements** The late Professor Helen Lester conducted the interviews and had oversight of their transcription. TW would like to acknowledge the support her NIHR Senior Investigator award and TW and EP the NIHR Biomedical Research Centre at the South London and Maudsley NHS Foundation Trust and King's College London.

**Contributors** MB, EP, SL, TW, NT, LMD and GD designed the randomised controlled trial. MM was the trial manager. MB developed the cognitive therapy for command hallucinations. MB and AM designed and led the training for cognitive therapy for command hallucinations and AM and EP led the supervision process and the rating of therapy tapes. LM and MM analysed the data and with MB wrote the first draft of the manuscript. LM and MM redrafted the manuscript. All authors read and approved the final manuscript. MB had full access to all the data in the study and had final responsibility for the decision to submit for publication.

**Funding** This work was supported by the UK Medical Research Council (G0500965) and the National Institute for Health Research. MB is part-funded by the National Institute for Health Research (NIHR) Collaborations for Leadership in Applied Health Research and Care West Midlands.

**Disclaimer** This paper presents independent research and the views expressed are those of the author(s) and not necessarily those of the NHS, the NIHR or the Department of Health and Social Care.

**Competing interests** None declared.

**Patient consent** Obtained.

**Ethics approval** NHS, The West Midlands Research Ethics Committee (number 06/MRE07/71).

**Provenance and peer review** Not commissioned; externally peer reviewed.

**Data sharing statement** Anonymised interview transcripts are available for further analysis under the University of Warwick data sharing policy (http://www2.warwick.ac.uk/services/ris/research_integrity/code_of_practice_and_policies/research_code_of_practice/datacollection_retention/research_data_mgt_policy). Please contact MB.

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
