## [Reviewer comments · BMJ Open]

ARTICLE DETAILS

TITLE (PROVISIONAL)	The COMMAND trial of cognitive therapy for harmful compliance with command hallucinations (CTCH): a qualitative study of acceptability and tolerability in the UK.
AUTHORS	Birchwood, Max; Mohan, Laura; Meaden, Alan; Tarrier, Nick; Lewis, Shon; Wykes, Til; Davies, Linda; Peters, Emmanuelle; Michail, Maria

VERSION 1 – REVIEW

REVIEWER	Emi Ikebuchi Teikyo University School of Medicine, Department of Psychiatry, Japan
REVIEW RETURNED	30-Mar-2018

GENERAL COMMENTS	This study added valuable aspects of effectiveness of cognitive therapy for harmful compliance with command hallucinations to the large randomized controlled study COMMAND trial (CTCH) by the same authors. This qualitative study adhered to COREQ protocol excellently, and findings are useful for clinical implication and implementation of cognitive behavioral therapy for psychosis. Explanation is needed: 25 participants of this study were selected randomly from 98 Participants who belonged to the intervention group in CTCH. The authors described in the RCT article of the Lancet 2014 that 7 discontinued intervention and 18 were lost at follow up. However all of 25 in this study had completed all of the sessions as part of the CTCH program, and had consented to be interviewed. How this happened? Modification is needed: “concerns with therapy” is valuable information for clinical implication. In discussion section, the authors commented “few dropped out”. The exact number of dropped out rate are needed. In addition, how to treat concerns such as “it did not make the voices go away” is important information for readers. Further explanation is warranted.
--

REVIEWER	Ryan Balzan Flinders University, Australia
REVIEW RETURNED	15-Apr-2018

GENERAL COMMENTS	The authors present some interesting qualitative findings on acceptability and tolerability from the COMMAND trial (CBT for command hallucinations or CTCH). The paper focuses on what participants found helpful and about the therapy, and some of their primary concerns. This type of research is particularly helpful to
---

	clinicians conducting CBTp/CTCH (e.g., targeting aspects of the therapy that clients find most useful), and may help to improve outcomes for clients by improving therapeutic alliance and minimising issues with therapy. The study was well conducted (e.g., interviewers were not directly involved with administering therapy) and the paper is well written (findings presented clearly). I only a few minor concerns:  - Details on CTCH were minimal. I understand that the focus of the paper was not reviewing the underlying theory of the therapy or its evidence-base, but some extra details in the Introduction or Method might help readers unfamiliar with the approach better interpret the results – e.g., average number of sessions, length/frequency of sessions, how the approach expands upon ‘regular’ CBTp (i.e., how does it “weaken and challenge beliefs about voices’ power”)? - One the listed aims of CTCH is to “reduce harmful compliance behavior and distress”. While the authors list that participants found “gaining control over the voices” and “challenging the power and omniscience of the voices” helpful, I did not see any specific mention of the therapy reducing distress or whether improvements translated into functional gains (e.g., return to work; improved social skills). Could the authors speak to this? - While a strength of the paper is that interviewers were not involved with therapy, I wonder if the interview format itself may have made responses more vulnerable to social desirability? Were there any self-reported measures of acceptability and tolerability used throughout the trial (e.g., at the end of each session)? It would have been good to see how such measures aligned with the interviews, particularly as opinions towards therapy could change over time. - Moreover, how long after therapy ended were the interviews conducted? Was this uniform across participants?
--	---

VERSION 1 – AUTHOR RESPONSE

Reviewer: 1

This study added valuable aspects of effectiveness of cognitive therapy for harmful compliance with command hallucinations to the large randomized controlled study COMMAND trial (CTCH) by the same authors. This qualitative study adhered to COREQ protocol excellently, and findings are useful for clinical implication and implementation of cognitive behavioral therapy for psychosis.

Explanation is needed: 25 participants of this study were selected randomly from 98 participants who belonged to the intervention group in CTCH. The authors described in the RCT article of the Lancet 2014 that 7 discontinued intervention and 18 were lost at follow up. However all of 25 in this study had completed all of the sessions as part of the CTCH program, and had consented to be interviewed. How this happened?

We clarify that 98 participants were randomised to the CTCH group. Of those, 7 discontinued the intervention and 18 were lost to follow up (total n=25). The remaining 73 participants completed the CTCH and 25 of those were selected randomly to take part in the qualitative interviews.

Modification is needed: “concerns with therapy” is valuable information for clinical implication. In discussion section, the authors commented “few dropped out”. The exact number of dropped out rate

are needed. In addition, how to treat concerns such as “it did not make the voices go away” is important information for readers. Further explanation is warranted.

We have added in the discussion section that 7 participants discontinued the intervention.

The core aim of the CTCH was to weaken and change beliefs about voices’ power, thus enabling the individual to resist or appease them thereby reducing harmful compliance behaviour and distress, regardless of changes in voice activity i.e. intensity and frequency of voices. Although this is something that participants acknowledged in their interviews, some of them were left somewhat disappointed that the therapy did not “get rid of the voices”.

Managing expectations of the therapy is crucial for therapeutic success and may influence the relationship between therapist and client (Ekberg et al, 2016; Greenberg et al, 2006). Therapists have a key role in communicating clearly at the outset -and throughout the therapy- the intended changes and benefits of the therapy. This will allow a shared understanding of how therapy will proceed, facilitating engagement and rapport. We have amended our discussion to include this key point.

Reviewer: 2

The authors present some interesting qualitative findings on acceptability and tolerability from the COMMAND trial (CBT for command hallucinations or CTCH). The paper focuses on what participants found helpful and about the therapy, and some of their primary concerns. This type of research is particularly helpful to clinicians conducting CBTp/CTCH (e.g., targeting aspects of the therapy that clients find most useful), and may help to improve outcomes for clients by improving therapeutic alliance and minimising issues with therapy. The study was well conducted (e.g., interviewers were not directly involved with administering therapy) and the paper is well written (findings presented clearly). I only have a few minor concerns:

- Details on CTCH were minimal. I understand that the focus of the paper was not reviewing the underlying theory of the therapy or its evidence-base, but some extra details in the Introduction or Method might help readers unfamiliar with the approach better interpret the results – e.g., average number of sessions, length/frequency of sessions, how the approach expands upon ‘regular’ CBTp (i.e., how does it “weaken and challenge beliefs about voices’ power”)?

We thank the reviewer for this comment. We have now amended the Introduction to include further details of the CTCH.

- One of the listed aims of CTCH is to “reduce harmful compliance behavior and distress”. While the authors list that participants found “gaining control over the voices” and “challenging the power and omniscience of the voices” helpful, I did not see any specific mention of the therapy reducing distress or whether improvements translated into functional gains (e.g., return to work; improved social skills). Could the authors speak to this?

Reducing distress linked to the voices and re-gaining control over one’s life was a recurrent theme emerging from the data. We provide here the following quotes to support this:

Referring to the gains of the therapy one participant said “It [referring to the therapy] got me out of the house...I’m working and you know, I can shake them off [referring to voices]”. Similarly, another participant mentioned “I am going out and about on my own now and I can go up town when I want to. Just started going back to the Villa matches”. Feeling less distressed by the voices following the therapy, this participant noted “I feel much better now...very strong and not stressed all the time. I don’t want the spirits to come back again. Always get better and do things you like to do”.

- While a strength of the paper is that interviewers were not involved with therapy, I wonder if the interview format itself may have made responses more vulnerable to social desirability? Were there

any self-reported measures of acceptability and tolerability used throughout the trial (e.g., at the end of each session)? It would have been good to see how such measures aligned with the interviews, particularly as opinions towards therapy could change over time.

- Moreover, how long after therapy ended were the interviews conducted? Was this uniform across participants?

Interviews about the acceptability of CTCH and user views about the active ingredients of the intervention took place at the end of the therapy – this was uniform across participants. Although interviewers were not involved with the therapy, we cannot rule out social desirability bias. We did not use any self-reported measures of acceptability and tolerability and we acknowledge that in the limitations section.

VERSION 2 – REVIEW

REVIEWER	Emi Ikebuchi Department of Psychiatry, Teikyo University School of Medicine, Japan
REVIEW RETURNED	05-May-2018
GENERAL COMMENTS	The manuscript was exactly modified according to reviewers' comments.
REVIEWER	Ryan Balzan Flinders University, Australia
REVIEW RETURNED	05-May-2018
GENERAL COMMENTS	The authors have addressed and/or clarified my comments in this revision.